# Effect of Cold and Hot Compounds Forming on Microstructures and Mechanical Properties in the Deformation Zone of Sharp-Edged High-Strength Steel Sections

**DOI:** 10.3390/ma16176001

**Published:** 2023-08-31

**Authors:** Wenqiu Yao, Jingtao Han, Chunjing Wu, Zhongqian Cao

**Affiliations:** 1Institute for Advanced Materials and Technology, University of Science and Technology Beijing, Beijing 100083, China; 2Guangzhou Sino Precision Steel Tube Industry Research Institute Co., Ltd., Guangzhou 511300, China

**Keywords:** cold and hot compound forming, roll forming, high-strength low-alloy structural steel

## Abstract

We researched the cold and hot composite-forming process, setting the forming speed at 2 m/min, the induction heating frequency at 40 KHz, and the induction current at 3000 A, and manufactured curtain wall steel with sharp corners. We analyzed the microstructure and mechanical properties of the deformation zone of the sharp-edged section using a tensile test, impact test analysis, metallographic observation (OM), fracture morphology observation, and electron backscatter diffraction (EBSD) analysis. The results show that the formed profile had a 96% reduction in the radius of the outer fillet and a 76% increase in the thickness of the corner compared to the pre-formed shape. The tensile strength increased by 3.6%, and the elongation after break increased by 13%. A forming temperature of 850 °C and forming deformation of 70% were determined as the optimum process parameters.

## 1. Introduction

Curtain wall keel sections are made from steel and aluminum. Aluminum is suitable for lightweight curtain walls [1,2,3], but its production consumes more energy [2,4,5] and offers poor fire resistance [6,7]. Steel can be processed through methods like cold bending and welding. Cold bending results in hollow profiles, which changes their mechanical properties and microstructure. This process increases their strength and hardness due to prestress but decreases their ductility and toughness [8]. The external corners of these profiles are rounded and not pointed [9,10]. High residual stress is observed at these corners [11,12,13], and both the yield stress and ultimate stress in the corner and adjacent areas are comparatively high [14,15]. In contrast, curtain wall keels produced by welding have sharp external corners and weld seams on the internal corners, leading to high residual stress at these corners. The welding-induced residual stress impacts the buckling performance of the curtain wall keel section when subjected to compressive loads, thereby reducing its ultimate axial strength [16,17,18,19,20,21]. Steel sections reign supreme when supporting considerable glass lengths, slender structures, and fire protection requirements [22,23]. Because of their light weight, permeability, and safety, they are preferred for their ability to meet these needs [24]. Typically, steel roll-formed sections are easy to produce and install and use less steel [22], but it may not be possible to achieve a shape with sharp outer rounded corners. Welding, on the other hand, enables one to create a more pointed shape, but this has drawbacks, such as welds at each corner joint, reduced dimensional accuracy in the case of extended members, and requiring straightening and grinding for optimal appearance [23].

Compared with the traditional process for welding steel plates, the hot and cold composite-forming technology has higher production efficiency because it can complete the desired shape in one go. Manufacturing costs can be reduced if the production speed is increased during the production process. Estimating costs at an early production stage allows companies to make optimal decisions about the production process [25].

Alloy sections have apparent technical advantages. Aluminum alloy (6063) has a tensile strength of 140 MPa and a shear strength of 81.2 MPa. In comparison, Q355B steel has a tensile strength of 540 MPa and a shear strength of 370 MPa, making steel more resistant to deformation and providing a more significant support capacity. Steel sections support larger glass areas than aluminum sections of the same cross-sectional shape, giving a greater free span and increased glass permeability. Additionally, the modulus of elasticity for Q355B steel is 208 GPa [26], almost three times that of aluminum alloy (6063) at 70 GPa, allowing steel members to be slimmer than aluminum alloys while meeting the deflection limits. In terms of fire protection, steel has better fire resistance than aluminum alloy because its melting point is 1500 °C, while aluminum alloy (6063) has a melting point of 650 °C [6]. Thus, steel can be used with fireproof glass to make fireproof curtain walls.

Now, thanks to advances in manufacturing technology, sharp-edged high-strength steel sections, which have replaced steel sections used for curtain wall keels, yield a shape with sharp outer rounded corners and guarantee accurate profile dimensions. However, the processing of high-strength steel can be affected by various problems, such as cracking [27,28], twisting [29,30], longitudinal bending [31,32], cut-off port deformation [33,34], area thickness thinning [35], large bending radii [36], residual stresses in the bending area, and hardening [33,37]. This paper proposes a novel forming technique to produce sharp-edged high-strength steel sections and counter these defects: cold and hot compound creation. This technique involves roll forming combined with electromagnetic induction heating technology for the local heating of the deformation zone. The flexibility of high-strength steel increases with the forming temperature [38,39,40], so introducing electromagnetic induction heating technology can lessen the quality and shape accuracy of the product in the subsequent forming process, thereby overcoming the defects caused by the roll forming and reducing the number of forming operations. This technique reduces the residual stress across the thickness of the curtain wall keel section [41], leading to a uniform grain structure, an enhanced toughness, and the elimination of residual stress. Grain refinement strengthens the material’s strength and toughness [42]. By refining the grain, the material’s ductility [43] and elongation rate [44] are enhanced, facilitating the creation of profiles with sharp corners. According to Han et al. [9], V-shaped high-strength steel sections with a thickness-to-bending radius ratio of less than two can be challenging to roll from without damage due to the low elongation of high-strength steel.

Yan [45] suggested using induction heating to study the warm roll forming of high-strength steel. Mehari [41] reviewed the stress–strain of induction heating on the roll-forming process of square and rectangular high-strength steel tubes. Peng [46] used induction heating to form square high-strength steel tubes with sharp corners at 650 °C. Wang [47] used induction heating to create square high-strength steel tubes with a forming temperature of 950 °C. The above scholars’ research is mainly focused on closed-end steel. The cold and hot composite-developing technology has not been used for open-end steel. This study is of great significance for developing and improving steel curtain wall keels. By adopting cold and hot composite-forming technology, the manufacturing of steel profiles with sharp outer rounded corners can be realized, thus promoting the progress and development of the construction industry. In the curtain wall industry, the use of sharp-edged steel profiles is crucial, as they can be used to support and connect the components of a curtain wall to ensure its stability and safety. Using hot and cold composite-forming technology to produce profiles with sharp external rounded corners can better meet the needs of curtain wall design while improving the overall quality and esthetics of the curtain wall. It can also provide more innovative solutions for the design and construction of curtain walls and open up new possibilities for optimizing the appearance and structure of buildings.

## 2. Materials and Methods

### 2.1. Materials

The material used in the experiments was high-strength low-alloy structural steel , grade Q355B, manufactured by Anyang Iron and Steel in Anyang, Henan, China, with a plate thickness of 3 mm. Table 1 shows the chemical composition of the steel, and the primary microstructures were ferrite (F) and pearlite (P) [48].

### 2.2. Experimental Setup and Methods

Figure 1 depicts the cold and hot composite-forming process. The forming equipment comprises four passes, each consisting of top rolls, bottom rolls, side rolls, and a core mold. The diameter of the upper roll is 300 mm; the diameter of the lower roll is 300 mm, and the diameter of the vertical roll is 160 mm. An induction heating coil was installed in the middle of the first and second passes. We used a forming speed of 2 m/min, an induction heating frequency of 40 KHz, and an induction current of 2000 A in the forming process, and the preformed steel section had a height of 160 mm, a width of 60 mm, an outer corner radius of R = 6 mm, and a corner thickness of T = 3 mm, as shown in Figure 2a. For the first through pass, the hole had a height of 159 mm, a width of 59 mm, and an outer corner radius of R = 5 mm, through the induction heating coil. For the second pass, the hole had a height of 159 mm, a width of 59 mm, and an outer corner radius of R = 5 mm. Next, through the induction heating coil, into the second pass, the hole had a height of 157.5 mm, a width of 57.5 mm, and an outer corner radius of R = 0.5 mm. Continuing to the third pass, the hole had a height of 156.8 mm, a width of 56.8 mm, and an outer corner radius of R = 0.2 mm. Continuing to the fourth pass, the hole had a height of 156.5 mm, a width of 56.5 mm, and an outer corner radius of R = 0.2 mm. To obtain the final profile, the formed section had a length of 156.5 mm, a width of 56.5 mm, an outer fillet radius of R = 0.2 mm, and a corner thickness of T = 5.3 mm, as per Figure 2b. The section after forming corresponded to a 96% reduction in the outer fillet radius and a 76% increase in the corner thickness from the performed section. We analyzed three cases based on the roll deformation amount in the second pass. The first case included 70% roll deformation (2.45 mm); the second case was 90% roll deformation (3.15 mm), and the third case was 100% roll deformation (3.5 mm). We selected the roll-forming temperatures to be 850 °C, 900 °C, and 950 °C for all three cases.

Depending on the deformation amount, the 1st roll deformation is 0 mm. Then, the 2nd roll deformations are 70%, 90%, and 100%. The 2nd deformation is a precision-forming operation. Afterward, the third pass is for finishing and shaping. Lastly, in the 4th pass, the section is shaped. Additionally, before the 1st pass, an induction heating coil was passed through. The corner of the area was heated by electromagnetic induction heating to temperatures of 850 °C, 900 °C, and 950 °C to decrease plasticity at the desired corner. As a result, the section can continue through the 2nd pass, 3rd pass, and 4th pass to obtain a sharp corner section.

### 2.3. Tensile and Impact Tests

Tensile specimens were taken from the direction of model rolling, as shown in Figure 3, and elongated to failure at a 1 mm/min rate at room temperature (25 °C) through a universal testing machine (Type CMT5205). Additionally, a Charpy pendulum impact test was performed using the GB-T229-2007 [49], using a Shanghai Laiyang Electric Technology Co., Ltd., JBS-300B pendulum impact tester in Shanghai, China, and a specimen size, as shown in Figure 4. These tests investigate the impact properties of the material.

### 2.4. Metallographic Organization Observation and Grain Size Determination

The metallographic organization of specimens obtained using different processes was observed and compared through metallographic microscopy. Wire cutting was used for the metallographic dimensioning, and then the surfaces of the samples were first hand-polished and then mechanically polished. Finally, the polished surface was rinsed with alcohol and then blown dry, followed by etching with a 4% nitric acid–alcohol solution. The microstructure of the formed specimens was analyzed using an MV5000 metallurgical microscope from Nanjing Jiangnan Yongxin Optics Co., Ltd., located in Nanjing, Jiangsu, China

First, we obtained a microscopic photograph of the sample through an MV5000 metallurgical microscope. With the help of Nano Measurer 1.2.5 software and the intercepting line method, we measured and recorded the particle size of the piece. To improve the accuracy of the measurement, we created statistics for the photographs of different regions of sample cel2 and finally obtained the average value.

### 2.5. Fracture Morphology Observation and EBSD

A Carl Zeiss scanning electron microscope (Oberkochen, Germany) (MERLIN Compact) was employed to observe the morphological aspects of the impact fracture. The sample was mounted on a stub with a graphite coating to achieve this, and it was subsequently exposed to vacuum pressure. The microscope was operated at 10 kV to guarantee the most accurate results.

The grain orientation, grain boundary characteristics, and crystallographic texture distribution were analyzed using electron backscatter diffraction (EBSD) measurement techniques. The surfaces of the samples were cleaned by mechanical polishing with 2000-grit sandpaper and then polished electrolytically with a solution of 10% perchloric acid + alcohol. An EDAX USA, Inc. Hikari XP electron backscatter diffractometer with a scanning step of 0.3 μm was used for EBSD testing.

## 3. Results

### 3.1. Analysis of Tensile and Impact Test Results

We chose a second-pass forming deformation of 90% to analyze the effects of different forming temperatures of 850 °C, 900 °C, and 950 °C on tensile properties. From Figure 5a, we found that the maximum tensile strength of the original material was 448.4 MPa, and the elongation after the break was 23.92%. An analysis revealed that when the forming temperature was 850 °C and the forming deformation was 90%, the material’s tensile strength increased to 567.4 MPa and the elongation after the break was 26.81%. The forming temperature at 900 °C and the 90% forming deformation created essentially a similar tensile strength and extension as the original material. When the forming temperature was 950 °C and the forming deformation was 90%, the material’s tensile strength was higher and the elongation after the break was lower than the original material. The increase in tensile strength of the material is due to the high solubility of C and N elements at high temperatures, which fully solidify into the matrix material and play a solid solution-strengthening role [50]. The decrease in material elongation is mainly due to the coarsening of the material grains due to high-temperature heating [51].

We chose the second-pass forming deformation of 70% and 100% to analyze the effect of a forming temperature of 850 °C on the tensile properties of the material. As seen in Figure 5b, at 850 °C and 70% deformation, the material achieved a tensile strength of 464.4 MPa and an elongation after fracture of 27.07%. The tensile strength of the formed material is 3.6% higher than that of the original material, and the extension after the break is 13% higher than that of the raw material. Similarly, a forming temperature of 850 °C and 100% deformation yielded a tensile strength of 448.52 MPa and an elongation after break of 23.64%, comparable to the original material. The results demonstrate that the material is austenitized at 850 °C, shifting from flattened grains to an equiaxed state and forming smaller polygonal ferrite [52]. After 70% deformation, the crystal size homogenizes, thus improving the material properties. Consequently, the 850 °C and 70% deformation process parameters yielded the best performance.

We evaluated the forming deformation of 90% at the second pass and the influence of developing temperatures of 850 °C, 900 °C, and 950 °C on the impact properties of the material. As illustrated in Figure 6a, the impact toughness of the specimen considerably rose when the forming temperature was 850 °C. Additionally, a slight increase was seen in the impact work when the forming temperature progressed to 900 °C and 950 °C. These results suggest that, with the temperature rise, there is a slight increase in impact work, with the best impact toughness at 950 °C and the lowest at 850 °C. The impact properties are poorer because the forming deformation of 90% leads to many dislocations within the material, especially within the partial unrecrystallized ferrite, resulting in the hardening of some areas of the material [53].

The decrease in impact work of the material at a forming temperature of 850 °C and 70% forming deformation is due to an increase in the [47] {001}<110> unfavorable weave resulting in a decrease in impact work, leading to a decline in the impact work. Studies have suggested [50] that {001}<110> weaving hurts material properties and increases material brittleness. The reduction in impact strength of materials with a forming temperature of 850 °C and 100% forming deformation is attributed to the considerable deformation causing a vast number of dislocations in the interior unrecrystallized areas of the material, making them hardened, and the origin of specimen cracks, consequently reducing the impact properties. When the forming temperature is 850 °C, the specimens’ yield strength, elongation, tensile strength, and impact work are more significant than the original material, thus yielding better overall performance.

### 3.2. Metallographic and Grain Size Analyses

Figure 7a shows the lateral structure of the original material specimen, consisting mainly of massive or striped ferrite, although the ferrite is relatively tiny. When the forming temperature is 850 °C and the forming deformation is 90%, part of the ferrite in the specimen is austenitic after cooling, forming equiaxed elliptical ferrite whose size is elongated along the rolling direction, as shown in Figure 7b. When the forming temperature is 900 °C and the forming deformation is 90%, the material is fully austenitic at 900 °C and forms equiaxed massive polygonal ferrite after cooling, with relatively homogeneous grains and arbitrary grain orientation, without any obvious co-orientation pattern, as illustrated in Figure 7c. When the forming temperature is 950 °C and the forming deformation is 90%, the ferrite grains become coarsened and slatted and their grain orientation is arbitrary, as shown in Figure 7d. As the forming deformation increases, the ferrite tissue morphology in the specimen will change; with a forming temperature of 850 °C and a deformation amount of 70%, the grain has a directional tendency, with larger ferrite elongated toward the rolling direction and smaller ferrite pieces scattered around, resulting in an overall equiaxed exemplary ferrite grain configuration, as shown in Figure 7e. Moreover, with a forming temperature of 850 °C and a deformation amount of 100%, the grain sizes were significantly reduced, forming long ferrite grains distributed along the rolling direction and more, as shown in Figure 7f. All these results collectively demonstrate that with increased forming deformation, the material grains tend to orientate in the rolling direction and become finer. As grain refinement has a more significant impact on improving the mechanical properties of steel [54], there will be an increase in the toughness and strength of the material and a reduction in grain size.

The size of the test sample grains was analyzed with Nano Measurer 1.2.5 software, combining it with a metallographic microscope to obtain metallographic pictures. Three hundred grain lines were measured for the short-axis length of the grains, obtaining the distribution ratio graph of the materials’ grain size. Figure 8 is the bar graph of the grain size distribution. Figure 8a shows that the original grains are mainly between 0 μm and 4 μm, with 80% of grains smaller than 4 μm. Less than 1% of grains have a size of more than 16 μm, displaying an apparent polarization in particle size. In Figure 8b, with the forming temperature of 850 °C, the particle size becomes larger overall, with most of them between 2 μm and 4 μm, the polarization of the particle size disappears, and the structure becomes uniform. When the forming temperature is 900 °C, the average grain size increases with the increase in the forming temperature, with most between 2 μm and 8 μm.

The number of grains larger than 10 μm significantly increased, while those smaller than 2 μm diminished. Particle size polarization is evident, indicating that with the increase in the austenitizing temperature, some grains grow more extensive and the structure becomes nonuniform. In Figure 8c, the material undergoes recrystallization with the 900 °C forming temperature and the grain becomes more significant than the 850 °C grain. There is no polarization of the particle size, and the structure is uniform. In Figure 8d, with the forming temperature of 950 °C, The number of large grains increases more and more, with some grains becoming larger. Comparing and analyzing this, Figure 9 plots a line graph of grain size distribution. 

As it turns out, when the forming temperature increases, the grain boundary migration rate of the metallic material increases. It can lead to a gradual merging of initially smaller grains to form larger grains, increasing grain size. As the forming temperature increases, the rate of grain boundary migration increases, which may lead to a gradual decrease in the difference between grain sizes. Smaller grains may gradually increase, while larger grains will shrink progressively under grain boundary migration, leading to homogenization of the grain size distribution. Overall, these phenomena can be attributed to the increase in the rate of grain boundary migration at high temperatures, which makes the change in grain size more pronounced.

### 3.3. Analysis of Fracture Morphology

As can be seen from Figure 10a,b, the specimen exhibits reduced impact resistance due to the increased dislocation density and consequent hardening. Figure 10c–f reveal that the specimens have superior impact toughness, owing to their forming temperatures of 900 °C and 950 °C, 90% forming deformation, widespread formation of challenging nests of uniform size, and sparsely distributed small construction platforms. However, higher temperatures tend to cause grain coarsening. Figure 10g,h indicate that fractures are governed mainly by challenging nests that are uniformly distributed and possess extensive and profound dimensions. Figure 10i,j show that the fracture is filled with small deconstruction fracture platforms connected by parabolically shaped challenge nests near the fracture, resulting in reduced impact performance. As it turns out, the plastic toughness is better at 70% of the forming deflection. At lower-forming deflections, the material undergoes more uniform plastic deformation, with a gradual increase in the dislocation density of the grains, but not to the extent of a high degree of accumulation. The material absorbs and disperses the impact energy better under impact loading, thus exhibiting better impact properties. The hardening of some areas at 100% percent of the forming deformation leads to deterioration of the properties, and the high-forming deformation leads to a sharp increase in the density of dislocations in some areas, which interferes with each other and builds up, resulting in a reduction in the plastic deformation capacity of the local area. It leads to the hardening of some regions’ formation of dislocation plug sets and micropores, reducing the material’s impact toughness.

Considering the results of the tensile and impact tests, the optimum process parameters were determined to be a forming temperature of 850 °C and a forming deflection of 70%.

### 3.4. Electron Backscatter Diffractometry (EBSD)

This experiment was analyzed using electron backscatter diffraction (EBSD) [55] to obtain a forming temperature of 850 °C and a forming deformation of 70% of the specimen grain orientation. The RD of the specimen was parallel to the *X*-axis of the carrier table, and the ND was similar to the *Z*-axis during the experiment. Figure 11 shows the specimens’ ODF, polar, and antipodal plots with 70% deformation measured by EBSD.

Figure 11 shows the ODF, the inverse polar plot of the EBSD-measured forming deformation, of a 70% specimen. The analysis reveals a large number of {110}<211> surface weaves found in the sample, with a maximum density of 5.62 in the {110}<211> weave. As it turns out, crystal orientation is related to deformation, and during the forming process, deformation leads to changes in direction within the crystal, resulting in a {110}<211> weave.

Parallel to the rolled surface of the trace of {111} faces was left. Additionally, Figure 12a signals that many <111> directions are parallel to the RD. Altogether, these results suggest that the weave of the specimens is primarily of the {110}<211> type.

Figure 13 shows a comparative graph of recrystallization statistics for different processes. According to references [56,57], dynamic recrystallization organizations, substructures, and deformed organizations are characterized by the use of grain orientation spreading. In Figure 13, blue illustrates the recrystallization area, yellow shows the substructure grain region, and red presents the grain deformation region. After being subjected to a forming temperature of 850 °C and a forming deformation of 70%, the material mainly contains recrystallized and deformed structures. Figure 13 shows 1.7% of the total recrystallized grains and 61.6% of the malformed grains in this specimen. The presence of a 36.7% substructure can lead to increased dislocation density forming, the creation of dislocation-dense regions, and the formation of substructures. As it turns out, at a forming temperature of 850 °C and a forming deflection of 70%, the temperature inside the material helps to induce recrystallization of some of the grains. The recrystallization process can lead to grain rearrangement and orientation changes. The large number of deformed regions and part of the number of substructures are due to the high deformation leading to a significant accumulation and movement of dislocations within the crystal. These deformation regions and substructures affect the mechanical properties of the material.

Changes in microstructure under the influence of deformation and recrystallization may lead to changes in the strength and toughness of the material. Recrystallization can help improve toughness by making the grain size of the material more uniform.

Large-angle grain boundaries can generally divide into normal large-angle grain boundaries and overlapping positions. These boundaries can improve the toughness of the material and inhibit crack propagation. A large number of grain boundaries with significant orientation differences boosts the dislocation density at subgrain edges, increases energy storage, and tends to lead to second-phase precipitation. When it comes to large-angle grain boundaries, CSL is usually the main focus of the study. Two grids with a joint subnet can be in a coincidence position under certain conditions. The parameter Ʃ is the ratio of the total overlapping position points, described as a Ʃ3 relationship when repeated at a three-frame position [58,59]. According to the CSL coincident position dot-matrix model, grain boundaries are classified into a large-angle, coincident dot-matrix, and random significant-angle grain boundaries. The low-energy limits are much like the low-energy recombination point boundaries, having less activity than the high-energy boundaries of the free grains and proving effective in strengthening material fracture toughness. When Ʃ has a value of 3, which happens when one of the two dot arrays is repeated at every three points, better atomic matches between the two grains give way to lower grain boundary energies. This, in turn, causes the elastic strain at the boundaries of the grains to decrease with the Ʃ value; for example, Figure 14 illustrates that Ʃ3 is the most abundant, taking up 3%. Experiments conducted at a forming temperature of 850 °C and forming deformation of 70% revealed that large-angle grain boundaries are mainly colattice Ʃ3 grain boundaries with low energy. These boundaries do not exhibit rolling segregation and are more stable and nonmigratory.

(1)The process parameters of a forming temperature of 850 °C and forming deformation of 70% help prevent material grain coarsening. This ultimately leads to a reduction in the internal dislocation density and improved material properties.(2)The selection of processing parameters of a forming temperature of 850 °C and a forming deformation of 70% can lead to a favorable {110} <211> weave, the {110} surface weave, mainly due to the increased forming deformation, which promotes the growth of recrystallized weaves of the material while deflecting the grains toward soft orientation, resulting in a larger angle of grain boundaries and improved tensile properties, leading to better overall material properties.(3)The forming temperature of 850 °C and a forming deformation of 70% can effectively guide the process of the production of high-strength sharp-angle steel. The material yielded has a greater yield strength, tensile strength, elongation, and impact work than the original material, and its overall performance is also improved. Furthermore, the outer radius of the corner of the section formed through cold and hot compound forming was reduced. In contrast, the corner thickness increased considerably compared to the preformed area.

Further research is planned following this paper:

In the cold and hot composite-forming process parameters, the 2 m/min developing speed is much lower than the production speed of the cold bending-forming process. Further research following this paper is planned to enhance the heating rate of the heating region, which will help to accelerate the scale production of hot and cold composite-forming technology applications.

## Figures and Tables

**Figure 1 materials-16-06001-f001:**
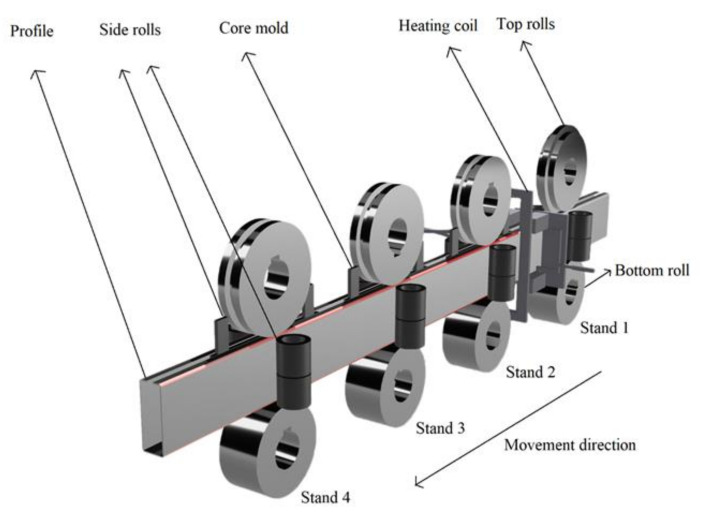
Cold and hot forming process.

**Figure 2 materials-16-06001-f002:**
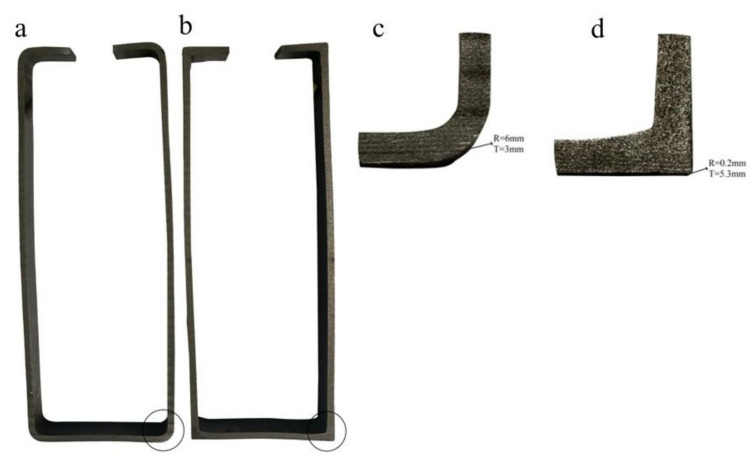
Comparing preformed and formed sections: (**a**) a preformed area; (**b**) a started section; (**c**) an enlarged corner of the preformed area; and (**d**) an enlarged corner of the formed site.

**Figure 3 materials-16-06001-f003:**
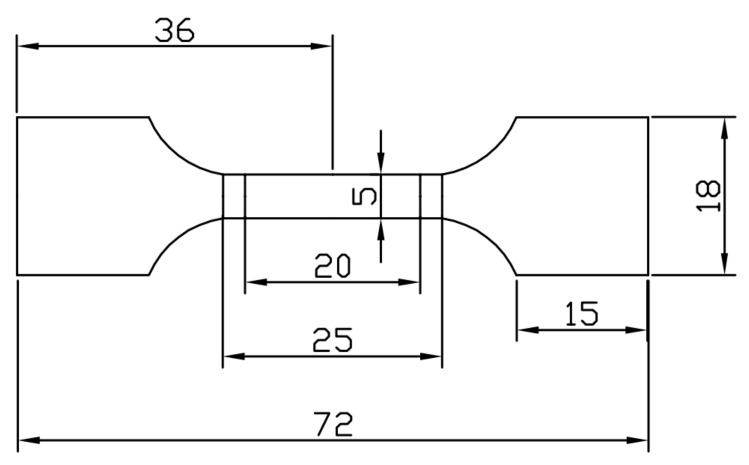
The dimensions of a tensile specimen (unit: mm).

**Figure 4 materials-16-06001-f004:**
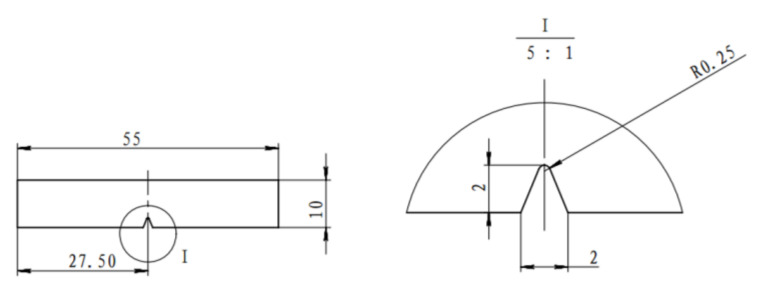
Dimensions of Charpy V-notch impact specimen (unit: mm).

**Figure 5 materials-16-06001-f005:**
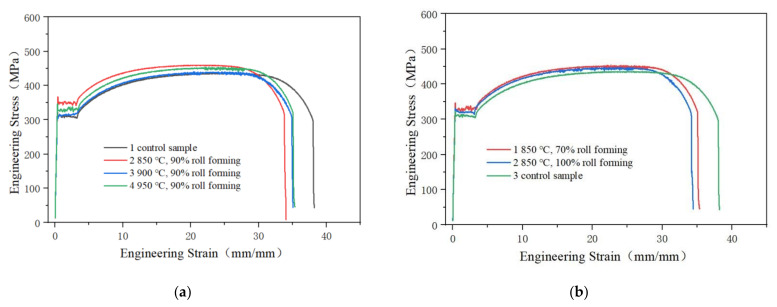
The tensile test results of specimens with different forming temperatures and deformation ratios: (**a**) the tensile test results for samples with different forming temperatures and a 90% forming deformation; (**b**) the tensile test results for models with various forming deformations at a forming temperature of 850 °C.

**Figure 6 materials-16-06001-f006:**
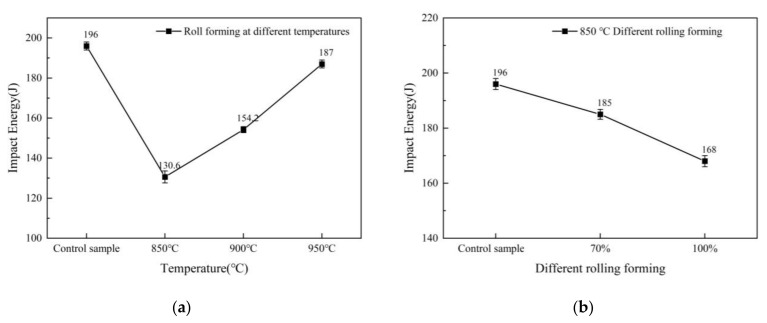
The impact test results of specimens with different forming temperatures and forming deformations: (**a**) the results for samples with varying forming temperatures and 90% forming deformations; (**b**) the results for specimens with different forming deformations at a forming temperature of 850 °C.

**Figure 7 materials-16-06001-f007:**
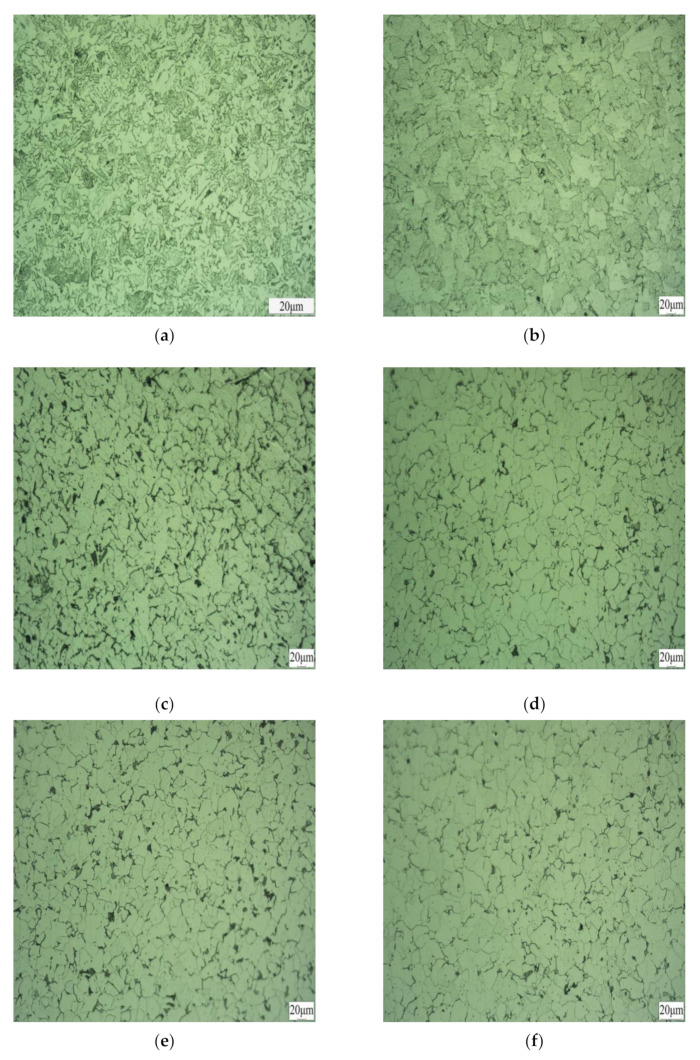
Photographs of the metallographic organization of specimens at various forming temperatures: (**a**) the original material; (**b**) forming temperature of 850 °C with 90% forming deformation; (**c**) forming temperature of 900 °C with 90% forming deformation; (**d**) forming temperature of 950 °C with 90% forming deformation; (**e**) forming temperature of 850 °C with 70% forming deformation; and (**f**) forming temperature of 850 °C with 100% forming deformation.

**Figure 8 materials-16-06001-f008:**
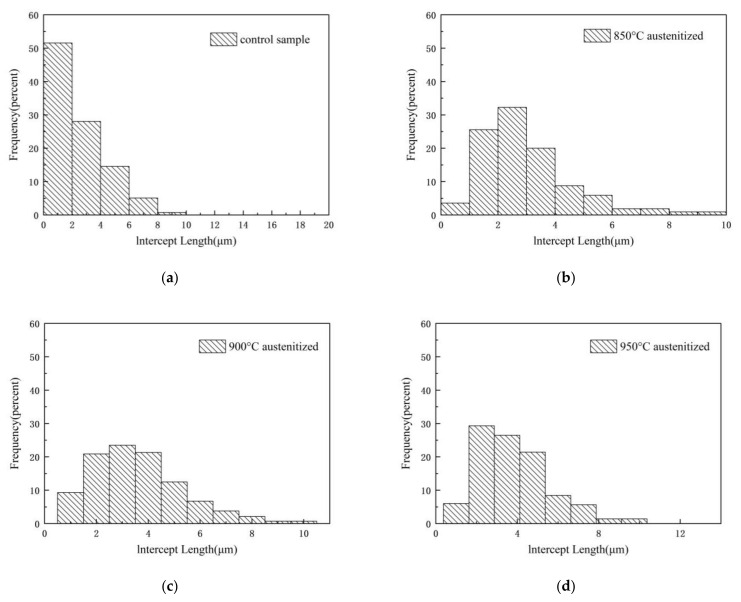
Histogram of grain size distribution: (**a**) histogram of grain size distribution for virgin material; (**b**) histogram of grain size distribution for forming temperature 850 °C and 70% deformation; (**c**) histogram of grain size distribution for forming temperature 900 °C and 70% deformation; and (**d**) histogram of grain size distribution for forming temperature 950 °C and 70% deformation.

**Figure 9 materials-16-06001-f009:**
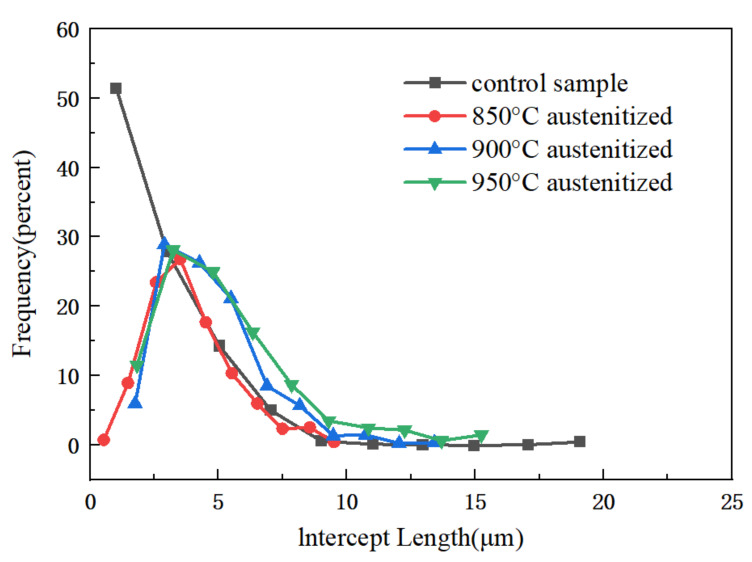
Folding line diagram of particle size distribution.

**Figure 10 materials-16-06001-f010:**
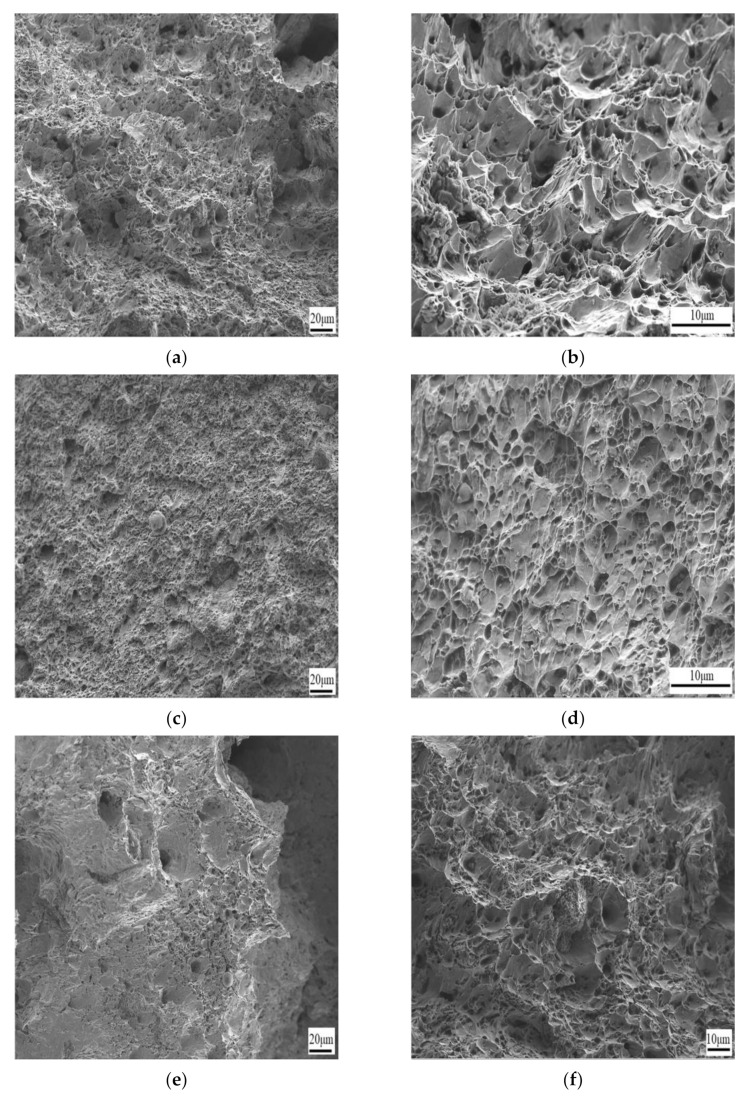
Scans of impact fractures of specimens at different forming temperatures and different forming deformations: (**a**,**b**) forming temperature 850 °C, 90% deformation; (**c**,**d**) forming temperature 900 °C, 90% deformation; (**e**,**f**) forming temperature 950 °C, 90% deformation; (**g**,**h**) forming temperature 850 °C, 70% deformation; and (**i**,**j**) forming temperature 850 °C, 100% deformation.

**Figure 11 materials-16-06001-f011:**
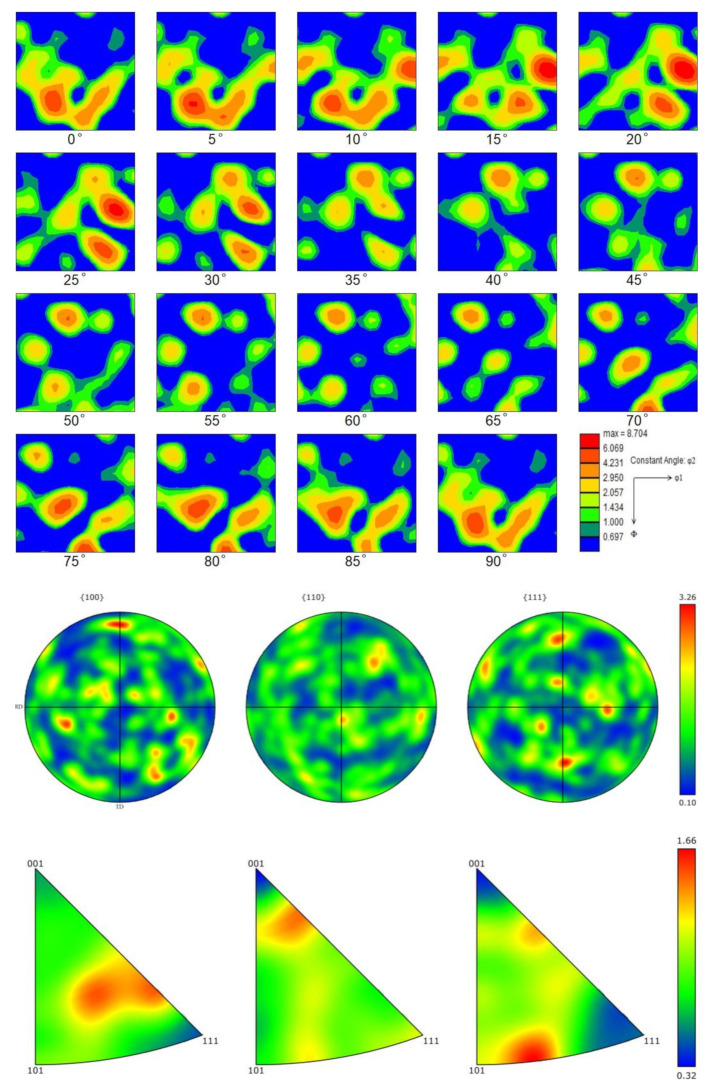
ODF, polar, and inverse polar diagrams of a sample with a forming temperature of 850 °C and a forming deformation of 70%.

**Figure 12 materials-16-06001-f012:**
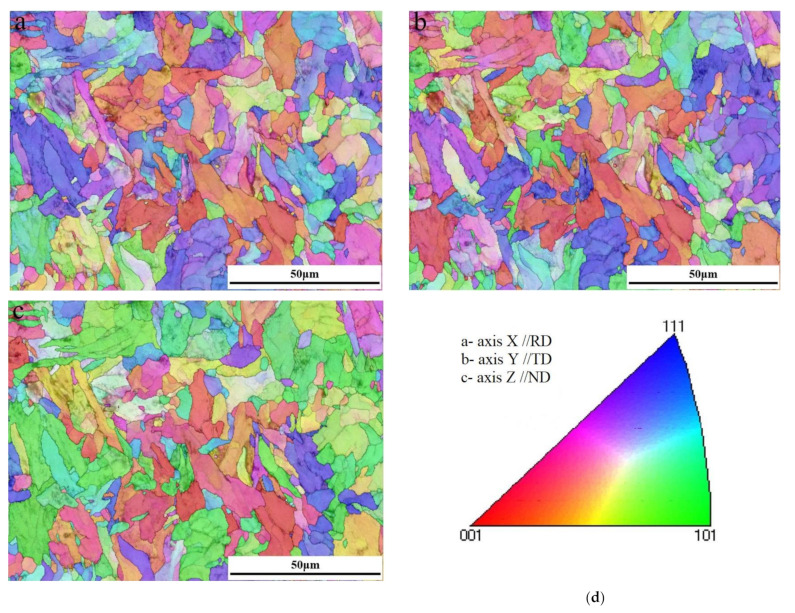
Schematic diagram of grain orientation of a specimen with a forming temperature of 850 °C and a forming deformation of 70%: (**a**) *X*-axis parallel to RD direction; (**b**) Y-axis parallel to TD direction; and (**c**) *Z*-axis parallel to ND direction. (**d**) correspondence between crystal plane orientation and color.

**Figure 13 materials-16-06001-f013:**
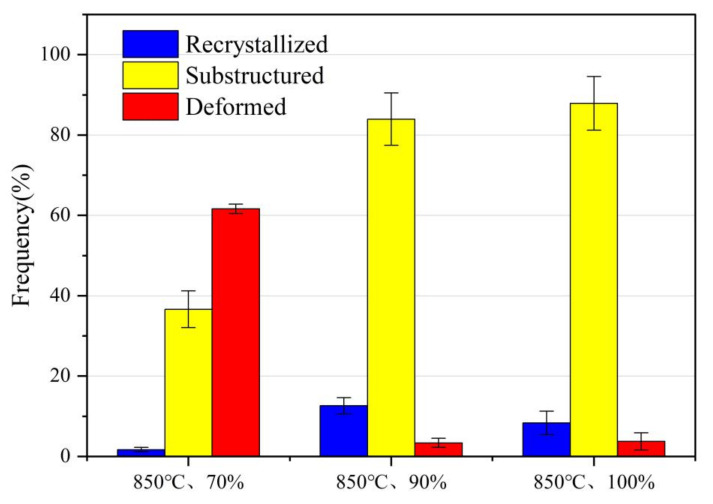
Comparison of recrystallization statistics for different processes.

**Figure 14 materials-16-06001-f014:**
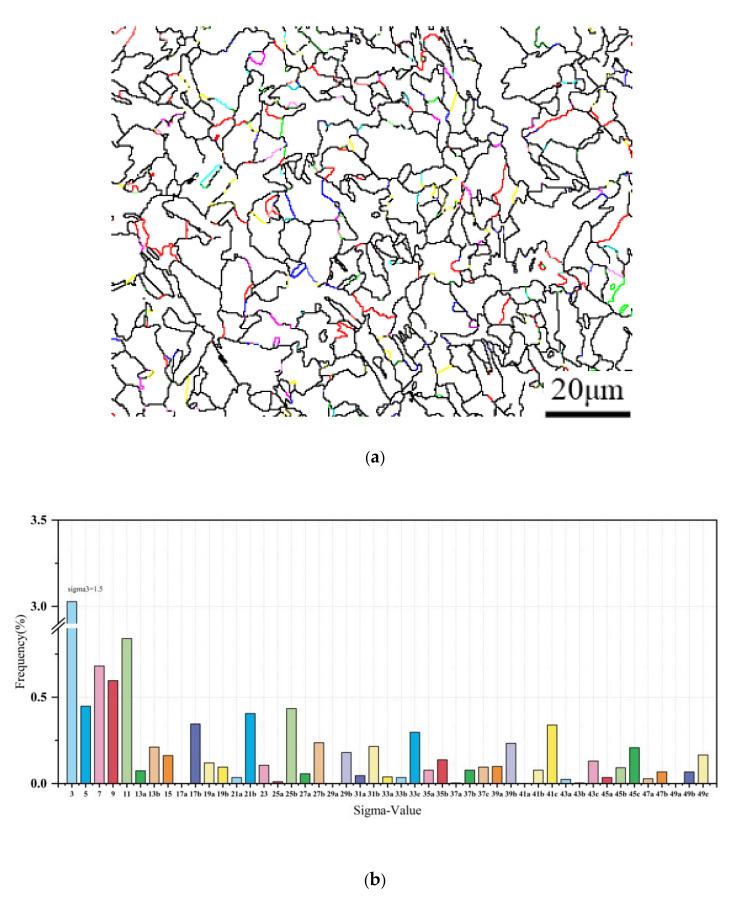
Depicts specimens formed at a temperature of 850 °C and a deformation rate of 70%. (**a**) shows the distribution at the overlapping positions. (**b**) statistics of the dot matrix at these overlapping positions.4. Conclusions.

**Table 1 materials-16-06001-t001:** Mass fraction of chemical composition of Q355B steel (wt.%).

C	Mn	Si	S	P
0.18	0.39	0.23	0.028	0.024

## Data Availability

Not applicable.

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
