# Peer review of "Effect of Cold and Hot Compounds Forming on Microstructures and Mechanical Properties in the Deformation Zone of Sharp-Edged High-Strength Steel Sections"

_materials, 2023, doi:10.3390/ma16176001_

Round 1

Reviewer 1 Report

1.Which is better cold or hot forming. Justify

2.Why there is an increase thickness of the corner after forming. 

3.How elongations helps in determining the material properties.

4.why forming temperature is 850 degrees is set as optimum. Why other values are not considered.

5. Production cost analysis is to be added in brief

6. Improve literatures. Include the following in the revised edition

https://doi.org/10.1016/j.matpr.2020.06.433

7. Improve grammar

Average

Reviewer 2 Report

In the summary part, no information is given about the process parameters.

In the introduction, similar studies in the literature were not evaluated in detail. Literature research should be added, the difference and novelty of this study should be explained.

Is the chemical composition given as a result of an analysis or is it catalog values?

Rolling speed, rolling width, roller radii, rolling roller material ?

Figures are generally small and of poor quality.

There are too many subheadings. This complicates the article for the reader.

Figures 13 and 14 are screenshots taken from the software. These data should be presented in a more understandable way by graphical form.

Information about the manufacturer of the Q355B steel should be given.

Although it is a good work in terms of the subject of the article, it is insufficient in terms of preparation and organization. Major revision opinion has been formed.

Reviewer 3 Report

Manuscript under consideration entitled „Effect of the cold and hot compound forming on microstructure and mechanical properties in the deformation zone of high-strength steel sharp-edged section” is an article that highlights important and current issues. I recommend the article for publication in Materials journal with the following comments:

1. In the introduction, it should be more strongly marked what is a scientific novelty in this article.

2. The methodology should provide information on the parameters of the individual tests. Generally a scientific article should contain all the information on the research methodology so that it is possible to repeat the experiments. In this work, the methodology is not clearly and fully described.

3. The SI notation for the unit of energy is "J" not "Joule".

4. The descriptions in some figures are illegible, like the scale in Fig. 11, the axis captions in Figs. 13, 14.

5. A more scientific discussion of the obtained research results should be carried out, which should include an in-depth analysis of the reasons for the occurrence of specific phenomena that are presented. The authors only state that a given phenomenon has been observed but a scientific discussion is required as to what was the cause of the observed phenomena. With no scientific discussion, this is more of a research report than a scientific publication

6. Discussion is too general. It is necessary to compare results obtained with results from references.

7. Discussion should contain plans for further research.

8. The literature review in the introduction should be significantly expanded, with the latest publications cited.

9. The formatting does not fully comply with the guidelines imposed by the journal.

10. Editing of English language and style required.

Editing of English language and style required.

Round 2

Reviewer 3 Report

The manuscript has been revised and significantly supplemented by the authors. In its current form, I recommend the article for publication in the journal Materials.

Minor editing of English language required.